# Effect of Chitosan/Thyme Oil Coating and UV-C on the Softening and Ripening of Postharvest Blueberry Fruits

**DOI:** 10.3390/foods11182795

**Published:** 2022-09-10

**Authors:** Haiyan Sun, Danqing Hao, Yun Tian, Yigang Huang, Yulin Wang, Gongwei Qin, Jinjin Pei, A. M. Abd El-Aty

**Affiliations:** 1National Engineering Research Center for Preservation of Agricultural Products in Qinba Area Preservation Workstation, Shaanxi Key Laboratory of Resource Biology, 2011 Qinling-Bashan Mountains Bioresources Comprehensive Development C. I. C, Qinba National Key Laboratory of Biological Resources and Ecological Environment, College of Biological Science and Engineering, Shaanxi University of Technology, Hanzhong 723000, China; 2Department of Pharmacology, Faculty of Veterinary Medicine, Cairo University, Giza 12211, Egypt; 3Department of Medical Pharmacology, Faculty of Medicine, Atatürk University, Erzurum 25240, Turkey

**Keywords:** blueberries, coating, UV-C, chitosan/thyme oil, softening, preservation

## Abstract

This study investigated the possible mechanism of softening and senescence of blueberry after harvest using chitosan/thyme oil coating combined with UV-C (short wave ultraviolet irradiation) treatment. On the 56th day of storage, the CBP, cellulose, and hemicellulose contents in the chitosan/thyme oil coating +UV-C-treated group were 1.41, 1.65, and 1.20 times higher than those in the control group. Compared with the control group, the activities of polygalacturonase (PG), pectin methylesterase (PME), β-glucosidase (β-Gal), and cellulose (Cx) were significantly reduced (*p* < 0.05) after chitosan/thyme oil coating +UV-C, and their maximum values decreased by 5.41 μg/h g, 5.40 U/g, 12.41 U/g, and 3.85 μg/h g, respectively. Moreover, chitosan/thyme oil coating combined with UV-C treatment inhibited the gene expression of PG, PME, Cx, and β-Gal and then regulated the decrease in PG, PME, Cx, and β-Gal activities, inhibited the degradation of cell wall polysaccharides, and delayed the softening and senescence of postharvest blueberries. The results showed that chitosan/thyme oil coating, UV-C, and chitosan/thyme oil coating + UV-C could significantly inhibit postharvest softening of blueberry; chitosan/thyme oil coating +UV-C had the best effect.

## 1. Introduction

Blueberry, belonging to the genus cranberry of the rhododendron family, is a dark blue fruit when mature and covered with white wax [1]. Most blueberries are nearly round; however, some varieties are oblate. North American native species have a delicate flesh and are delicious, sour, and sweaty when mature [2]. Blueberries have high nutritional value with many functions [3,4,5], such as antioxidant and antiaging properties, enhancing immunity, protecting eyesight, antitumor activity, lowering blood lipids, and improving memory [6,7,8,9]. Blueberries have become popular with improving living standards and awareness of healthy diets. The market potential is enormous; however, fresh blueberries tend to wilt after harvest, which reduces their edible and nutritional value. This severely restricts the global blueberry market. Postharvest storage and preservation technology has become an urgent problem for developing the blueberry industry [10].

Many studies have shown that chitosan-based composite coatings can delay the decline in blueberry firmness and decay rate, maintain the contents of vitamin C, total sugar and anthocyanins, control the activities of related softening enzymes, and improve storage quality [11,12,13]. Currently, chitosan, clove oil, quinoa protein, potassium sorbate and polylysine have been developed as coating-based materials for preserving blueberries [12]. Investigations have shown that they can positively impact blueberries’ physiological quality and extend their storage life. Thyme essential oil is a natural plant extract showing antimicrobial activity. Studies using thyme oil combined with coating treatment showed that they could significantly prolong the storage period and inhibit the declining quality of fresh sweet basil leaves, mango, and freshly cut apples [14,15,16]. There are no relevant reports regarding the application of thyme essential oil in preserving the postharvest quality of fresh blueberries. Using ultraviolet rays of appropriate wavelength, UV-C destroys the molecular structure of deoxyribonucleic acid (DNA) or ribonucleic acid (RNA) in microbial cells, resulting in the death of growing cells [17]. In this context, Villagra et al. found that UV-C treatment can delay the blueberries’ quality decline during storage and improve their antioxidant activity [18]. Furthermore, Zhou et al. found that UV-C could effectively inhibit the growth of microorganisms, maintain quality, and enhance the antioxidant activity of blueberries during postharvest storage [19]. On this occasion, combining chitosan composite and UV-C can effectively improve the storage quality and prolong the shelf life of blueberries. However, molecular gene-level studies have not been reported, and softening aging mechanisms are unclear.

In this study, we prepared a coating liquid with thyme essential oil and chitosan as the primary raw materials, and UV-C was used to keep blueberries fresh. The effects of different treatments on cell wall polysaccharides, related degrading enzyme activities, and gene expression during storage were also explored. Finally, we clarified the mechanism of chitosan/thyme oil combined with UV-C treatment on blueberry postharvest softening and aging.

## 2. Materials and Methods

### 2.1. Materials

Blueberries were picked from the blueberry base in Chenggu County, Hanzhong City, Shaanxi Province; the variety is “emerald”. Blueberries of the same size and maturity with no mechanical damage or diseases were selected. Fruits were placed in an incubator at 4 °C immediately after picking and shipped back to the laboratory for processing on the same day.

### 2.2. Methods

#### 2.2.1. Treatment Applied to Blueberries

(1) Preparation of chitosan/thyme oil coating solution: Nine grams of chitosan was dissolved in 200 mL of 1% (*v*/*v*) acetic acid. The samples were then placed in a water bath for 25 min at 55 °C. After adding 10 g gelatin, the samples were heated in a water bath until complete dissolution. Next, glycerol (30% chitosan, *v*/*w*) was added, followed by ultrasonic treatment until complete dissolution; 11 mL thyme essential oil and 11 mL Tween (R) 80 were mixed under ultrasonication. After that, 1% (*v*/*v*) acetic acid solution was added to a final volume of 1000 mL.

(2) UV-C treatment: The irradiation dose was 3 kJ/m^2^, and the UV lamp (vitamin D conversion box) was independently developed by Shaanxi University of Technology; 130 cm × 55 cm × 153 cm, cuboid, upper part sealed (90 cm), lower part open (63 cm)). The lamp (TUVPL-L36 W/4P, PHILIPS) was installed 30 cm from the top and was stable for 30 min after turning it on. The irradiation intensity under the UV lamp was determined by an ultraviolet irradiator (ST512uvc, SENTAR), and the irradiation time was calculated as 308 s according to the formula


T = 100 × irradiation dose/irradiation intensity.



(3) Blueberry preservation treatment. The experiment was divided into four groups: I. Control group (CK): no treatment. II. Chitosan/thyme oil coating group; (T): blueberries were soaked in the pre-prepared coating solution for 30 s and then removed and placed on a smooth and clean gauze (to avoid adhesion, the surface of the blueberries was covered with a layer of uniform and transparent film and dried naturally). III. Irradiation group (Z): Twenty blueberries were placed 60 cm away from the UV lamp, irradiated on one side for 154 s, and then subjected to continuous irradiation on the other side for 154 s. IV. Chitosan/thyme oil coating +UV-C group; (T+Z): the blueberries were soaked in the pre-prepared coating solution for 30 s and then placed on smooth and clean gauze to dry and irradiate. The specific irradiation operation and time were the same as those in treatment 3. After the above treatment, the blueberries were divided into crisper boxes. Each box contained 45 blueberries, labeled and weighed, and stored at a low temperature of 4 °C ± 0.5 °C. Samples were taken every 8 days, and each treatment was repeated three times.

#### 2.2.2. Index Measurement

##### Determination of Cell Wall Polysaccharides

(1) Determination of crude cell wall material content

The method of weighing was slightly modified [20] and the material content of coarse cell walls was expressed as a percentage.
(1)$$\mathrm{Crude}\;\mathrm{cell}\;\mathrm{wall}\;\mathrm{matter}\;\mathrm{content}\;(\%)=\frac{\mathrm{dry}\;\mathrm{crude}\;\mathrm{cell}\;\mathrm{wall}\;\mathrm{matter}\;\mathrm{content}}{\mathrm{weight}\;\mathrm{of}\;\mathrm{fresh}\;\mathrm{blueberries}}\times 100\%$$

(2) Determination of water-soluble pectin content

Water-soluble pectin (WSP) was determined using the carbazole colorimetric method and expressed as the percentage of galacturonic acid generated. The determination, extraction, and standard curve of galacturonic acid production were determined according to Kyriakidis et al., with minor modifications [21]. The standard curve of galacturonic acid was y = 0.0064x + 0.0177, *R^2^* = 0.9998. The water-soluble pectin content was calculated according to Formula (2).
(2)$$\mathrm{water}-\mathrm{soluble}\;\mathrm{pectin}\;\mathrm{content}\;(\%)=\frac{M\times \frac{V}{v}}{m}\times 100\%$$
where M is the mass of galacturonic acid (UG) in the standard curve;

V—Total volume of sample extract (mL)

v—Volume of the extracted liquid (mL)

m—Sample mass (g)

(3) Determination of ionic pectin content

Based on the above tests, 10 mL of 50 mM acetic acid-sodium acetate buffer solution (including 2 mM EDTA) with pH = 5.5 was added and placed in a water bath for 1.5 h at 50 °C. After cooling to room temperature, centrifugation was performed at 4200× *g* for 15 min. The supernatant containing ion-binding pectin was obtained. The determination method, calculation formula and standard curve of ionic pectin (ISP) are the same as in formula (2).

(4) Determination of covalent pectin content

Based on the above tests, 10 mL of 50 mM Na_2_CO_3_ solution (including 2 mM EDTA) was added to a water bath for 1.5 h at 50 °C, cooled to room temperature, and centrifuged at 4200× *g* for 15 min to obtain the covalent pectin supernatant. The covalent pectin (CBP) determination method, calculation formula, and standard curve are the same as stated in formula (2).

(5) Determination of hemicellulose content

Based on the above tests, 10 mL of 4 mM NaOH (including 100 M NaBH_4_) was added and placed in a water bath for 1.5 h at 50 °C, cooled to room temperature, and centrifuged at 4200× *g* for 15 min to obtain the supernatant containing hemicellulose.

The hemicellulose content was determined based on the method of Wang et al. [22]. The standard curve y = 13.689x + 0.0009, *R²* = 0.9993 was constructed using standard glucose. The hemicellulose content was calculated according to Formula (3).
(3)$$\mathrm{Hemicellulose}\;\mathrm{content}\;(\%)=\frac{m^{\prime }\times V\times N}{V_{s}\times m\times 10}\times 100\%$$
where m′-- is the mass of glucose (μg) in the standard curve;

V—Total volume of sample extract (mL);

N—Dilution ratio of sample extract, 1;

V_s_—The volume of liquid extracted from the samples (mL);

m—Sample mass (g)

(6) Determination of cellulose content

The remaining materials from the above tests were centrifuged and the residue was cellulose material, which was determined using the weighing method and repeated three times. Represented by %, calculated according to Formula (4).
(4)$$\mathrm{Cellulose}\;\mathrm{content}\;(\%)=\frac{\mathrm{dried}\;\mathrm{weight}}{\mathrm{crude}\;\mathrm{cell}\;\mathrm{wall}\;\mathrm{material}\;\mathrm{content}}\times 100\%$$

##### Determination of Cell Wall Degrading Enzyme Activity

(1) Extraction and determination of the activity of polygalacturonase

DNS colorimetry was used to extract and determine the activity of polygalacturonase (PG) [23]. A standard curve was generated with glucose y = 0.3725x − 0.0074, *R²* = 0.9993. PG activity was expressed as the mass of galacturonic acid produced by catalytic hydrolysis of polygalacturonic acid at 37 °C per gram of blueberry tissue sample (fresh weight) per hour, calculated according to Formula (5):(5)$$\mathrm{PG}\;\mathrm{activity}(\mu g/h\cdot g)=\frac{m^{\prime }\times V\times 1000}{v\times t\times m}$$
where m’ is the mass of glucose (mg) obtained from the standard curve;

V—Total volume of sample extract (mL);

v—Volume (mL) of the extracted liquid from the sample taken at the time of determination;

t—Enzymatic reaction time (h);

m—Sample mass (g)

(2) Extraction and determination of pectin methylesterase activity

The extraction and activity of PME were determined using NaOH titration [24]. The amount of enzyme required to consume 1 mmol NaOH per gram of fresh blueberry sample per min was set as 1 unit of enzyme activity (U), and the result was expressed as U/g min.
(6)$$\mathrm{PME}\;\mathrm{activity}\;(U/g\cdot \min)=\frac{(V-V_{0})\times c}{n\times t\times m}$$
where V is the total volume of extracted liquid consumed by the enzyme solution (mL);

V_0_—NaOH volume consumed by blank control (mL);

c—NaOH concentration, 0.05 mmol/L (h);

n—1;

t—reaction time (min);

m—Sample mass (g)

(3) Extraction and determination of cellulase activity

DNS colorimetry was used for extraction and determination of cellulase (Cx) activity [23]. The extraction method for the Cx enzyme solution was the same as the standard curve (1), calculated according to Formula (7).
(7)$$\mathrm{Cx}\;\mathrm{activity}(\mu g/h\cdot g)=\frac{m^{\prime }\times V\times 1000}{v\times t\times m}$$
where m’ is the mass of glucose (mg) obtained from the standard curve;

V—Total volume of sample extract (mL);

v—Volume (mL) of the extracted liquid from the sample taken at the time of determination;

t—Enzymatic reaction time (h);

m—Sample mass (g)

(4) Extraction and determination of β-galactosidase activity

The extraction and determination of the activity of β-galactosidase (β-Gal) were performed according to Trainotti [24]. The absorbance of each tube was measured at 405 nm. The standard curve y = 0.0061x − 0.0068, *R²* = 0.9991 was drawn with *p*-nitrophenol. The production of 1 μmol *p*-nitrophenol per gram fresh weight per min was defined as 1 enzyme activity unit (U).

##### Expression Analysis of Cell Wall-Degrading Enzyme-Related Genes

Eukaryotic mRNA sequencing was based on the Illumina NovaSeq 6000 sequencing platform, and all mRNA transcribed from fresh blueberry fruits was sequenced. The Illumina TruSeqTM RNA Sample Prep Kit was used for library construction in sequencing experiments [25,26,27].

### 2.3. Data Analysis

The experiments were replicated three times. Origin 2018 software was used to draw and differentiate the data, and IBM SPSS 25.0 statistics software (Armonk, NY, USA) was used to conduct variance analysis for each result.

## 3. Results and Discussion

### 3.1. Effects of Different Treatments on the Content of the Crude Cell Wall in Blueberry

The crude cell wall material of blueberries decreased during the whole storage period (Figure 1). Hydrolysis and degradation occur under the action of degrading enzymes such as PG, Cx, and PME, resulting in a decrease in the content of crude cell wall substances of blueberry. During the whole storage period, the content of the crude cell wall in the CK group was significantly lower than that in the other treated groups, except that it was slightly higher than that in the T group on the 16th day. On day 56, the contents of the crude cell wall in the T+Z group, T group, Z group, and CK group were 1.68%, 1.53%, 1.51%, and 1.36%, respectively, which were significantly higher than those in the CK group (*p* < 0.05). The effect of the T+Z group was the most obvious. All three treatments effectively reduced the content of crude cell walls in blueberry, and chitosan/thyme oil coating +UV-C treatment had the best effect. In this way, Deng et al. declared that the crude extract of blueberry cell walls gradually decrease during storage. Cold storage could effectively inhibit the decline of the blueberry cell wall crude extract and prolong the shelf life [28]. Yan et al. stated that the storage quality of raspberries could be effectively improved by reducing the material loss of cell walls [29]. Sinha et al. also reported that proper sample preservation could effectively inhibit cell wall metabolism in pears, maintain their structural integrity, and improve their storage quality [30]^.^ This may be because the chitosan/thyme oil coating forms a protective film on the surface of blueberry, which inhibits metabolic activities during storage, reduces the activity of polysaccharide-degrading enzymes in the cell wall, and effectively inhibits the loss of crude cell wall substances in blueberry. Thyme oil and UV-C can restrain the growth of the microorganisms on the surface, effectively controlling corruption and inhibiting the decline in blueberry coarse cell wall material content. Therefore, the three treatments can effectively reduce the cell walls of blueberry coarse material loss during storage, and chitosan/thyme oil coating combined with UV-C had the best effect.

### 3.2. Effects of Different Treatments on Pectin Content

Figure 2 shows the changes in blueberry pectin during the whole storage period. There is a large amount of insoluble propectin in the primary cell wall and mesocolium of immature fruit and vegetable tissues, which gradually separates from other substances to form water-soluble pectin as the fruit matures. Figure 2a shows that the WSP content of blueberries in the CK and T groups first increased and then decreased during storage. They all reached peak values of 1.31% and 0.99% on day 32. After 32 days, the WSP content of all groups decreased gradually. In the first 48 days, the changing trend of blueberry in the T+Z and Z groups was almost the same, but there was no significant difference in the growth rate (*p* < 0.05). From 48 day to 56 day, the WSP content of blueberry in group Z decreased by 0.067% but increased by 0.031% in group T+Z. The WSP content in the three treatment groups was significantly lower than that in the CK group during storage. The three treatments significantly inhibited the increase in WSP content (*p* < 0.05). ISP refers to soluble pectin bound by an ionic bond, and its change rule is shown in Figure 2b. ISP shows an overall trend of rising first and then declining. The results of the difference analysis showed that the ISP content of blueberry in the Z group was significantly lower than that in the CK group (*p* < 0.05), and there was no significant difference in ISP content among the other groups (*p* < 0.05). The three treatments had little effect on the changes in ISP content in blueberry. CBP refers to the small molecule pectin bound by a covalent bond, a crucial component reflecting the integrity of the cell wall. The content of CBP represents the degree of demethylation of protopectin in tissues. The content of covalent pectin generally decreases with the aging of fruits. Figure 2c shows that with the extension of storage time, the CBP content in each group decreased gradually. The CBP content of blueberry in the T+Z group was significantly higher than that in the other three groups (*p* < 0.05), and the CBP content in groups Z and T was significantly higher than that in group CK (*p* < 0.05). In the research of Wang et al., deacidification promoted the softening of blueberry fruit, increased the content of soluble pectin and reduced the content of water-insoluble pectin, which is consistent with this experiment [31]. Hanbo Wang et al. and Ji Yaru et al. also reported that during storage, under the action of related enzymes, covalent pectin was gradually degraded into water-soluble pectin, the cell wall structure was destroyed, and the quality of blueberries decreased [32,33]. Wang et al. suggested that kiwifruit experienced rapid softening and quality changes after harvest [31]. They found that the water-insoluble pectin content decreased during storage, and the water-soluble pectin increased, which was consistent with the results of this test. Then, all three treatments effectively inhibited the decrease in CBP content in blueberry, and chitosan/thyme oil coating +UV-C treatment had the best effect. During storage, CBP is degraded into soluble WSP under the action of pectinase, which causes a decrease in CBP content and an increase in WSP content. At the later stage of storage, the WSP content decreased, which may be caused by the continuous consumption of nutrients in blueberries and the consumption of WSP as a respiratory substrate. At this time, the degradation rate of CBP and ISP was lower than the consumption rate of WSP, resulting in the overall trend of WSP content increasing first and then decreasing. The results showed that the three treatments could significantly inhibit CBP conversion into WSP but had little effect on the change in ISP content during storage.

### 3.3. Effects of Different Treatments on Hemicellulose and Cellulose Contents in Blueberries

Figure 3a shows the change in hemicellulose content. Twenty-four days before storage, the hemicellulose content in each group increased gradually. In the CK group < Z group and < T group < T+Z group, the peak values reached 0.59%, 0.62%, 0.70% and 0.72% on day 24, respectively. After 24 days, the hemicellulose content in each group began to decrease. The hemicellulose content was ranked as CK group < Z group < T group < T group < T+Z group during the whole storage period. The hemicellulose contents in the CK, Z, and T groups and the T+Z group were 0.21%, 0.38%, 0.39%, and 0.45%, respectively, on day 56. Analysis showed that the hemicellulose content in the three treatment groups was significantly higher than that in the CK group (*p* < 0.05). Changes in cellulose content in each group during storage are shown in Figure 3b. The cellulose content of blueberries in all groups decreased gradually. The cellulose content of blueberries in the CK group decreased the most, followed by the Z group, T group, and T+Z group. On day 56, the cellulose content in the CK, Z, T, and T+Z groups was 44.67%, 48.89%, 51.96%, and 53.82%, respectively. According to the above analysis, chitosan/thyme oil coating, UV-C and chitosan/thyme oil coating +UV-C treatments can effectively inhibit the decrease in cellulose and hemicellulose contents in blueberries during storage, and the effect of chitosan/thyme oil +UV-C treatment is better than that of the two treatments alone. In the study of Hangjun et al., the contents of cellulose and hemicellulose gradually decreased with the extension of storage time [34]. Liu’s study found that fruit hardness decreased during postharvest cold storage and decreased cellulose and hemicellulose contents [35]. Therefore, inhibiting the degradation of blueberry cellulose and hemicellulose can effectively inhibit the decline in blueberry quality during storage. In strawberries, effective storage methods significantly inhibited the degradation of cellulose and hemifibroin during storage and prolonged the shelf life [36]. Wang et al. found that hemicellulose and cellulose loss in persimmon fruits was significantly inhibited after preservation treatment [37].

### 3.4. Effects of Different Treatments on the PG Activity of Blueberry

PG activity first decreased, then increased and then declined, as shown in Figure 4 below. The results show that PG enzyme activity first increased and then decreased during storage. This result is slightly different from the test results. This may be due to the good quality of the blueberry fruits in the early storage stage, and PG activity was inhibited under low-temperature storage. In the middle of storage, the stress of low temperature on PG activity decreased, and the enzyme activity increased. At the later storage stage, various metabolic activities changed the pH environment of blueberry fruits, and PG activity decreased. During storage, the PG activity peaks of the CK, T, Z, and T+Z groups were 77.62 μg/h g, 76.30 μg/h g, 77.37 μg/h g, and 72.21 μg/h g, respectively. Compared with the control group, the maximum enzyme activity in the three treatment groups was decreased by 1.32 μg/h g, 0.25 μg/h g, and 5.41 μg/h g, respectively. After 40 days, only the PG activity in the T+Z group was significantly lower than that in the CK group (*p* < 0.05). During the whole storage period, chitosan/thyme oil coating and UV-C had little effect on the PG enzyme activity of blueberry (*p* < 0.05), but the PG activity of blueberries was significantly inhibited after chitosan/thyme oil coating +UV-C treatment (*p* < 0.05). In a study by Sinath Chea et al., β-aminobutyric acid treatment inhibited PG enzyme activity during blueberry storage and alleviated postharvest deterioration of “Lanfeng” high cluster blueberry fruits during cold storage [38]. Megha et al. used a chitosan coating combined with pomegranate peel extract to treat pear fruits for cold storage and found that the coating reduced PG activity during cold storage of pears [39]. In a study on the storage and preservation of apricots, Cui et al. declared that storage at near freezing temperatures maintained the PG activity of apricots at a low level, delayed the degradation of pectin components, and protected the structure of the plant cell wall [40]. A study on strawberries also found that cold storage treatment could inhibit PG enzyme activity and control strawberry softening [36]. Ethanol vapor also inhibited PG enzyme activity and cell wall degradation of blueberries [33]. Therefore, inhibition of PG enzyme activity can regulate the degradation of the blueberry cell wall.

### 3.5. Effects of Different Treatments on the PME Activity of Blueberries

The PME activity in all groups increased and decreased 24 day before storage (Figure 5). The difference analysis showed that the PME activity in groups Z and T+Z was significantly lower than that in group CK (*p* < 0.05), but there was no significant difference between the T group and CK group (*p S*< 0.05). From 24 day to 32 day, PME activity increased rapidly in each group, and the maximum PME activity of the CK group, T group, T+Z group and Z group was 26.43 ± 1.28 U/g, 21.02 ± 1.11 U/g and 24.89 ± 1.10 U/g, respectively. After 32 days, the PME enzyme activity of each group began to decrease, and the PME enzyme activity of each treatment group was significantly lower than that of the CK group (*p* < 0.05). In other words, the three treatments effectively inhibited the activity of the PME enzyme in blueberry during middle and late storage. The effect of PME on the aging and softening of blueberry is mainly through the production of the PG enzyme-substrate, which then affects the pectin substance in the blueberry cell wall. In this study, the changing trend of PME at the later stage of storage was similar to that of PG; both decreased. In this study, the activity of the PME enzyme was different from that reported by Siyao et al. in the early storage stage [41]. This may be because the storage temperature in this experiment was 4 °C, which was significantly lower than the storage temperature studied by Siyao et al., resulting in a slight fluctuation of PME enzyme activity in the early stage of storage. However, Ji et al. concluded that decreasing PME enzyme activity is conducive to inhibiting blueberry softening in research on blueberry storage and preservation [33]. Haobo et al. also reported that effective preservation methods could inhibit the activities of blueberry PME and other enzymes from improving the storage quality of blueberries [32]. Coletta et al. suggested that effective preservation treatment can inhibit grape PME enzyme activity and prolong shelf life in grape-related research [42].

### 3.6. Effects of Different Treatments on the Cx Activity of Blueberries

Figure 6 shows the changes in Cx activity in blueberries during storage. The activity of Cx first decreased, then increased, and then decreased in all groups. On the 8th day of storage, the Cx activity of blueberries in the CK, T, Z, and T+Z groups decreased to 54.45 μg/h g, 52.39 μg/h g, 51.60 μg/h g, and 48.54 μg/h g, respectively. The activity of Cx increased rapidly from 8 day to 16 day and reached peak values of 78.17 μg/h g, 77.65 μg/h g, 73.78 μg/h g, and 74.32 μg/h g on 16 day. From 16 day to 56 day, the Cx activity of blueberries in each group decreased gradually. During the whole storage period, the Cx activity of blueberries in the T and T+Z groups was significantly lower than that in the CK group (*p* < 0.05), but there was no significant difference in Cx activity between group Z and group CK (*p* < 0.05). That is, chitosan/thyme oil coating and chitosan/thyme oil coating +UV-C treatment can effectively inhibit the Cx activity of blueberries, but UV-C treatment has little effect on the Cx activity of blueberries during storage. The results showed that the coating treatment significantly affected the cellulase activity of fruit. Studies on ethanol steam preservation of postharvest blueberries also showed that inhibition of cellulase activity was conducive to maintaining the integrity of the blueberry cell wall structure [33]. Saleem et al. also inhibited strawberry fruit softening by reducing the activity of cell wall-degrading enzymes such as Cx with an edible coating containing chitosan-based ascorbic acid [43]. In a study by Sinha et al., the Cx enzyme activity of pear fruit decreased significantly after a salicylic acid (SA)-enriched beeswax (BW) composite coating [30]. The results are similar to those of this test.

### 3.7. Effects of Different Treatments on β-Gal Activity in Blueberry

During storage, the β-Gal activity fluctuated widely. It increased rapidly from 0 d to 8 d and then decreased rapidly(Figure 7). At 16 d, the β-Gal activity of the CK, T, Z, and T+Z groups decreased to the lowest values of 28.83 ± 0 U/g, 28.18 ± 1.91 U/g, 27.25 ± 2.02 U/g and 28.72 ± 1.20 U/g. The activity of β-Gal increased slightly from 16 day to 24 day and then decreased slowly and gradually increased after 32 day. Studies have shown that β-Gal is closely related to the softening of blueberries after rubbing and is directly proportional to the softening degree of blueberries [33]. In this experiment, the β-Gal activity of blueberries stored in the early stage decreased rapidly on days 8–16. It may be that the fruit used in this experiment were eight ripening blueberries. Blueberries gradually mature, and their quality increases in the early storage stage. The activities of cell wall-degrading enzymes such as β-Gal decreased, while the activities of related enzymes increased gradually with the extension of storage time. Analysis showed that chitosan/thyme oil coating treatment and UV-C treatment had no significant effect on β-Gal enzyme activity during the whole storage period (*p* > 0.05), but β-Gal activity was significantly inhibited in the T+Z group (*p* < 0.05); that is, only chitosan/thyme oil coating treatment +UV-C treatment could significantly inhibit the activity of β-Gal in blueberry. Li et al. also showed that appropriate treatment could reduce the cost of fresh-keeping in passion fruit β-Gal enzyme activity, delay the degradation of cell walls during fruit senescence, and improve the postharvest storage quality of passion fruit [44]. In grapes, the alternative temperature in the postharvest cooling treatment of grapes improves β-Galr activity and reduces fruit hardness.

### 3.8. Effects of Different Treatments on the Expression of Polysaccharide-Degrading Enzyme-related Genes in the Blueberry Cell Wall

According to the preliminary test, among the three treatments, the chitosan/thyme oil coating +UV-C treatment had the best effect on inhibiting the degradation of cell wall polysaccharides and related enzyme activities during blueberry storage (Figure 8). After a comprehensive analysis, we selected blueberries stored for 8 day and 24 day in the CK group (a1, a2) and chitosan/thyme sesame oil film + UV-C group (b1, b2) for eukaryotic transcriptome sequencing. We expect to reveal the regulatory mechanism of chitosan/silme coating + UV-C treatment on blueberry softening senescence at the molecular level. The results are shown in Figure 8 below. In the figure, the abscissa is the group, the ordinate is the differential gene expression amount, and the different colored squares in the figure represent different genes.

The expression of polysaccharide degrading enzyme genes in group a was significantly higher than that in group b (*p* < 0.05), and chitosan/thyme oil coating +UV-C treatment can significantly inhibit the expression of cell wall polysaccharide degrading enzyme genes, which is consistent with the conclusion that chitosan/thyme oil coating +UV-C treatment can significantly inhibit the activity of cell wall polysaccharide degrading enzymes. A total of 26, 6, 13 and 8 genes related to PG, Cx, PME, and β-Gal activity were screened. Compared to the control group, the chitosan/thyme oil coating +UV-C treatment significantly reduced the PG-associated genes VaccDscaff9-Augustus-gene-276.20,

VaccDscaff44-Augustus-gene-16.23, and VaccDscaff43-Augustus-gene-7.29, and the functional annotations of these three genes were related to PG enzyme activity and cell wall polysaccharide content. Cx activity is related to the following genes: VaccDscaff28-Augustus-gene-24.25, VaccDscaff44-snap-gene-15.43, VaccDscaff49-Augustus-gene-27.20, and VaccDscaff20-Augustus-gene-353.15 were significantly downregulated after chitosan/thyme oil coating +UV-C treatment, and the functional annotations of these four genes were involved in the regulation of Cx enzyme activity. VaccDscaff11-processed-gene-261.20, VaccDscaff140-Augustus-gene-0.30, VaccDscaff37-Augustus-gene-187.25, VaccDscaff39-Augustus-gene-130.23, and VaccDscaff46-Augustus-gene-144.33 are involved in regulating PME enzyme activity and changing cell wall structure. The expression levels of tUS-gene-130.23 and VACCdSCAFF46-Augustus gene-144.33 in the chitosan/thyme oil coating +UV-C group were significantly lower than those in the control group. No gene related to β-Gal activity was significantly downregulated in the chitosan/thyme oil coating +UV-C treatment group. In a study by Siyao Wang et al., ethylene treatment exacerbated the softening of blueberry fruit by promoting cell wall degradation via stimulation of PE, PG, and β-gal enzyme activities and *VcPE* and *VcPG* expression [41]. In strawberry, chitosan coating-treated fruit showed a reduction in the content of water-soluble pectin, coincident with significant inhibition of the activities of PG and pectin PME and the expression of *FaPG1* and *FaPME1* relative to the controls [36]. Viviana Martins et al. induced grapes with calcium spray before harvest. It was found that the expression of the cellulose synthase family *CesA3* was not affected by exogenous Ca, while the expression of the polygalacturonase coding genes PG1 and PG2 was downregulated [45]. Li Yaling et al. treated apricot fruit with salicylic acid; the results showed that salicylic acid treatment inhibited the expression of PaPG1 and PaPME1 genes, which were closely related to the activity of related enzymes, and these two genes were closely related to fruit softening [46].

## 4. Conclusions

After harvest, blueberries are prone to softening and quality decline, which seriously affects blueberry market sales. The degradation of cell wall polysaccharides is an important cause of blueberry fruit softening. During storage, under cell wall polysaccharide degrading enzymes, the water-insoluble pectin was transformed into water-soluble pectin, hemicellulose and cellulose were degraded and exfoliated, the intercellular layer dissolved, and primary and secondary walls were destroyed, resulting in the collapse of fruit cell wall reticular structure and fruit softening. The results showed that chitosan/thyme oil coating, UV-C and chitosan/thyme oil coating +UV-C could significantly inhibit CBP conversion into WSP and decrease cellulose and hemicellulose contents in blueberries during storage. However, each treatment had little effect on the ISP content in blueberries after harvest. Among the three treatments, the chitosan/thyme oil coating +UV-C treatment had the best effect on maintaining the polysaccharide content in the blueberry cell wall, and the contents of CBP, cellulose and hemicellulose were 1.41, 1.65 and 1.20 times higher than those of the control group on the 56th day of storage. The degradation of fruit cell wall polysaccharides occurs under the regulation of related polysaccharide-degrading enzymes and enzyme genes. Compared with the control group, the activities of PG, PME, β-Gal, and Cx were significantly inhibited by chitosan/thyme oil coating +UV-C treatment (*p* < 0.05), and their maximum values were lower than those of the control group. However, chitosan/thyme oil coating and UV-C treatment did not significantly inhibit the activities of PG, PME and β-Gal in blueberry. Illumina sequencing results showed that chitosan/thyme oil +UV-C treatment could significantly inhibit the gene expression intensity of PG, PME, Cx and β-Gal in blueberry. The changes were consistent with the changes in enzyme activity. In conclusion, chitosan/thyme oil +UV-C treatment can inhibit the activities of PG, PME, Cx and β-Gal by controlling the expression intensity of related enzyme genes and reducing the loss of polysaccharides in the blueberry cell wall, thus maintaining the integrity of blueberry cell wall tissue structure and inhibiting postharvest softening and senescence of blueberry.

## Figures and Tables

**Figure 1 foods-11-02795-f001:**
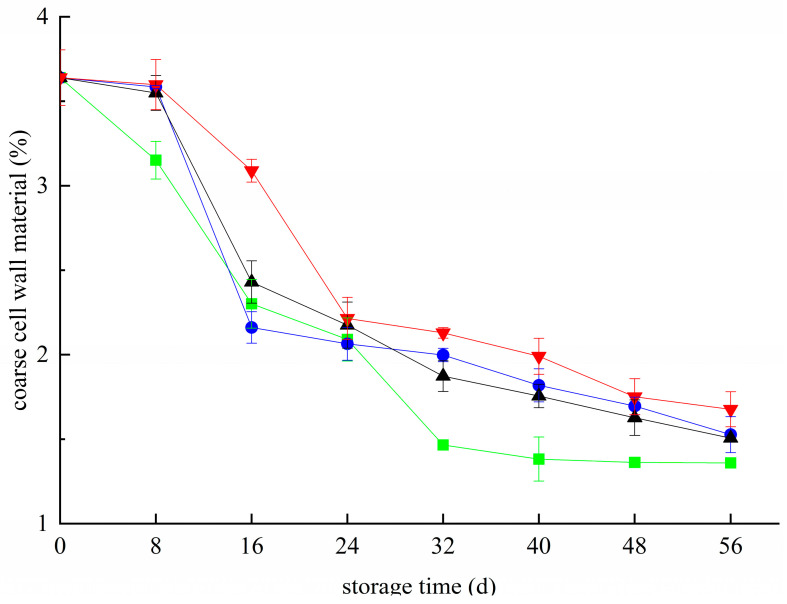
Effect of coarse cell wall material in blueberry under different treatments. ■: control; ●: T; ▲: Z: ▼: T plus Z.

**Figure 2 foods-11-02795-f002:**
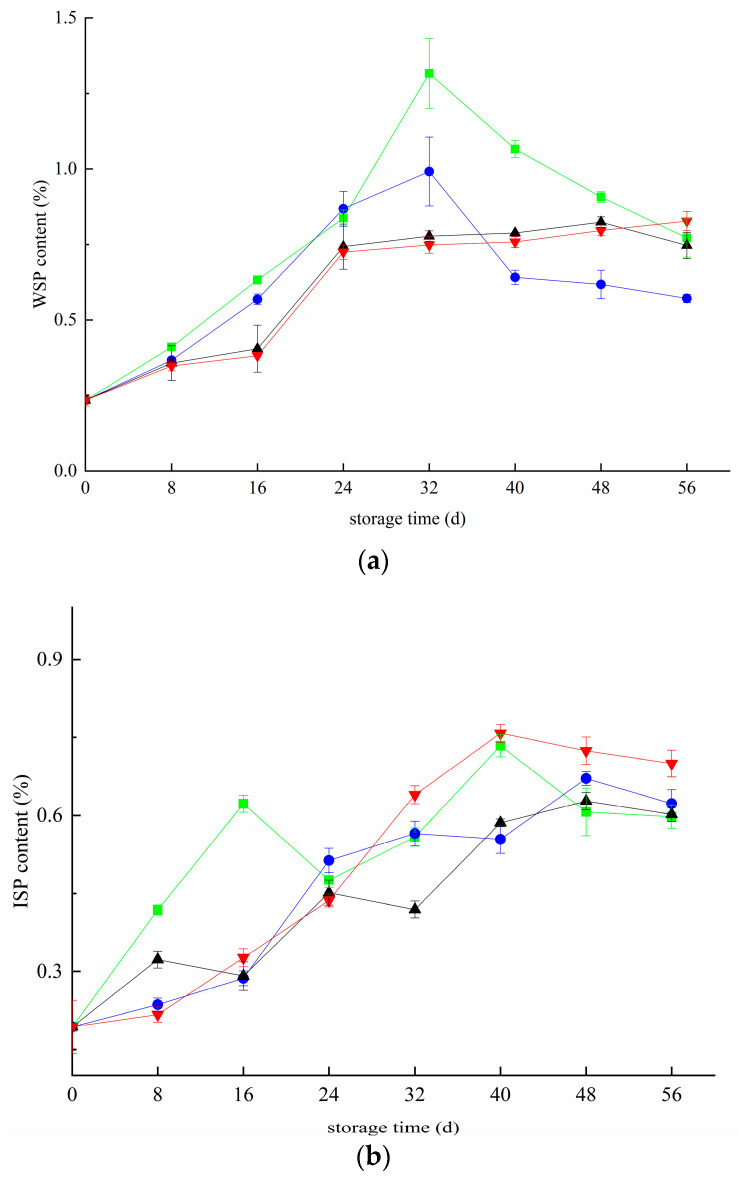
Effect of pectin substance in blueberry under different treatments (**a**–**c**). a: Changes in water-soluble pectin content in blueberry; b: Changes in ionic pectin content in blueberry; c: Changes in covalent pectin content in blueberries. ■: control; ●: T; ▲: Z: ▼: T plus Z.

**Figure 3 foods-11-02795-f003:**
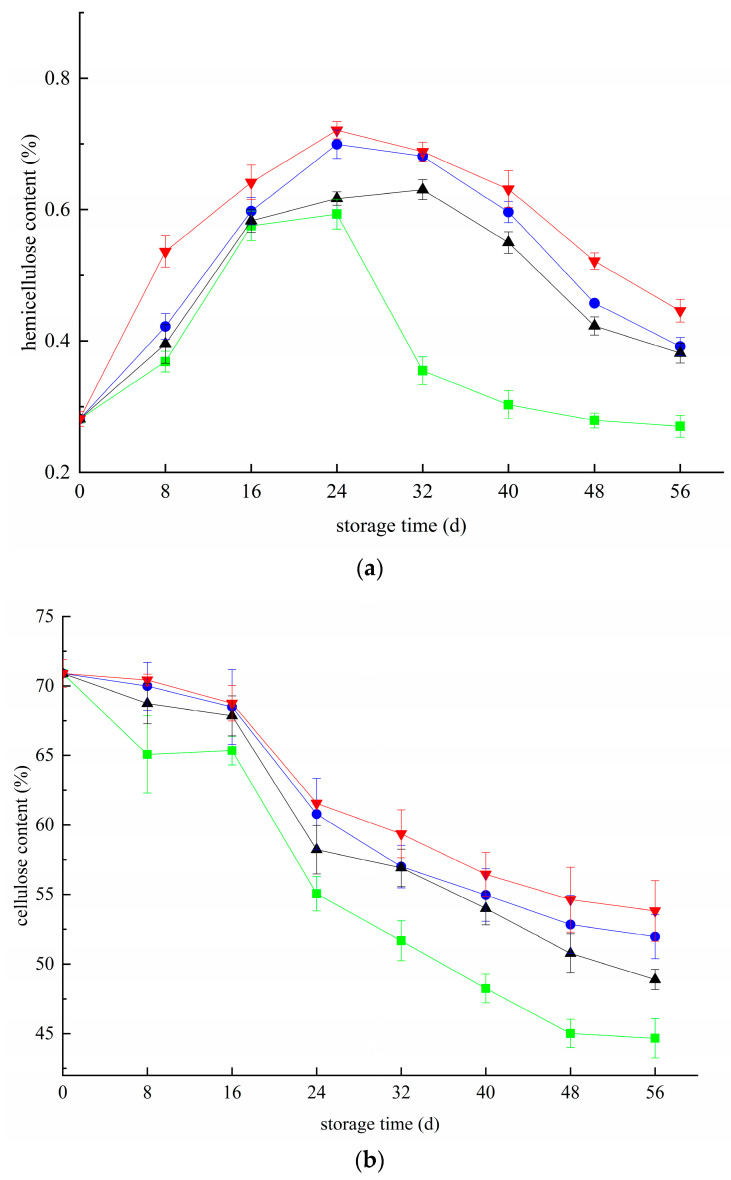
Effect of hemicellulose and cellulose in blueberry under different treatments (**a**,**b**). (**a**)**:** Change plot of hemicellulose content in blueberry; (**b**) Change plot of cellulose content in blueberry. ■: control; ●: T; ▲: Z: ▼: T plus Z.

**Figure 4 foods-11-02795-f004:**
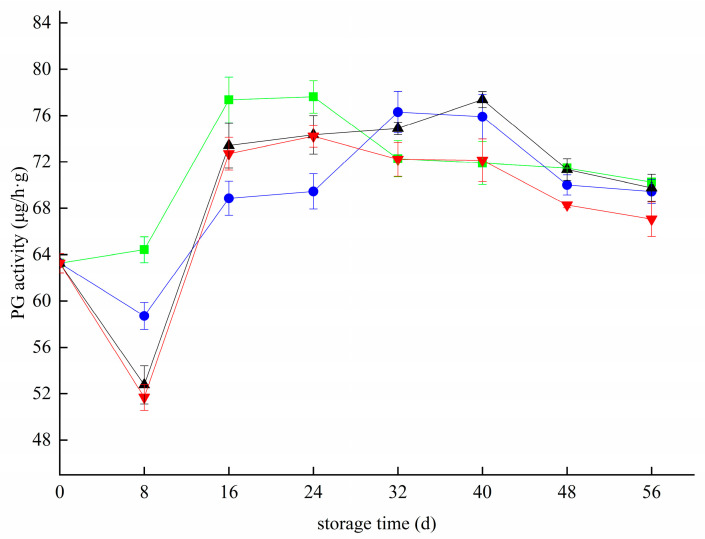
Effect of polygalacturonase activity in blueberry under different treatments. ■: control; ●: T; ▲: Z: ▼: T plus Z.

**Figure 5 foods-11-02795-f005:**
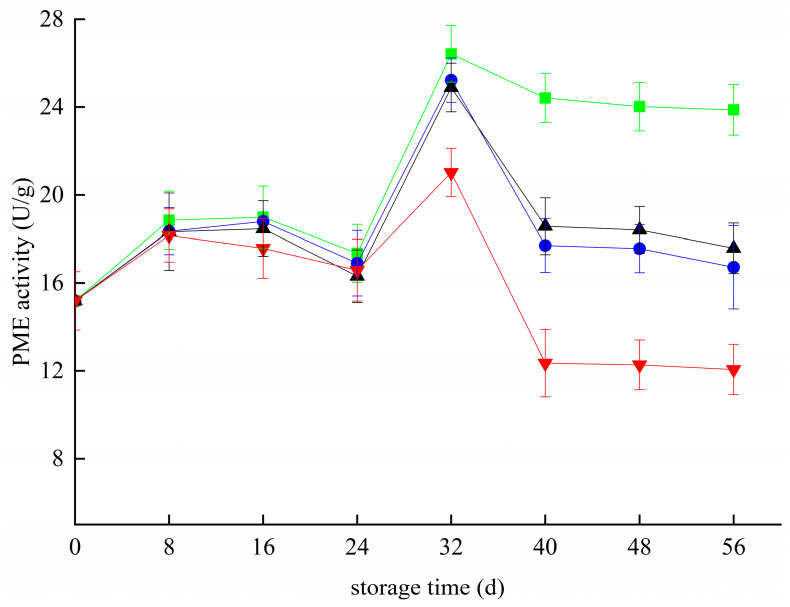
Effect of pectin methylesterase activity in blueberry under different treatments. ■: control; ●: T; ▲: Z: ▼: T plus Z.

**Figure 6 foods-11-02795-f006:**
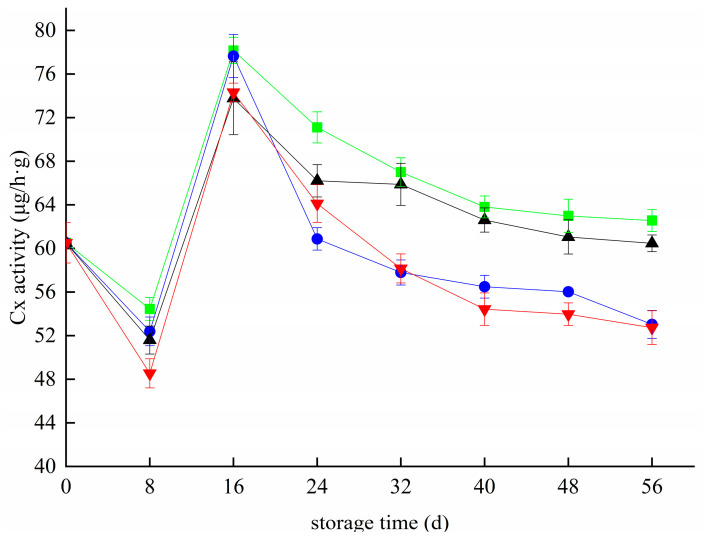
Effect of cellulase activity in blueberry under different treatments. ■: control; ●: T; ▲: Z: ▼: T plus Z.

**Figure 7 foods-11-02795-f007:**
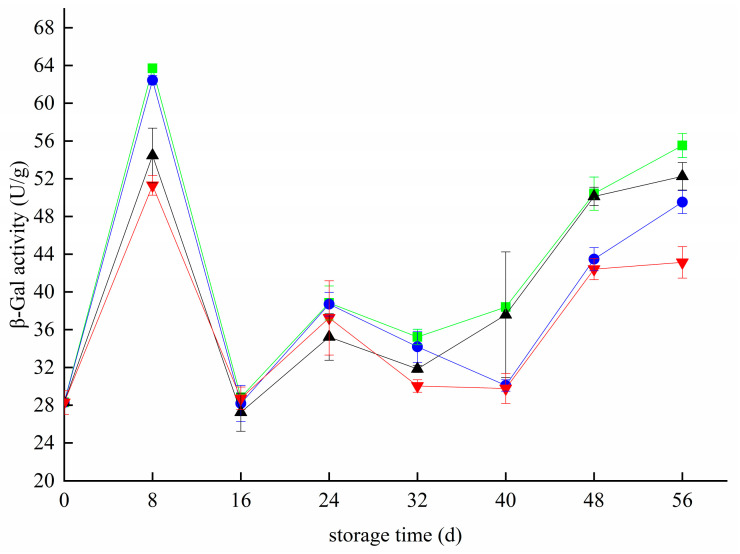
Effect of β-galactosidase activity in blueberry under different treatments. ■: control; ●: T; ▲: Z: ▼: T plus Z.

**Figure 8 foods-11-02795-f008:**
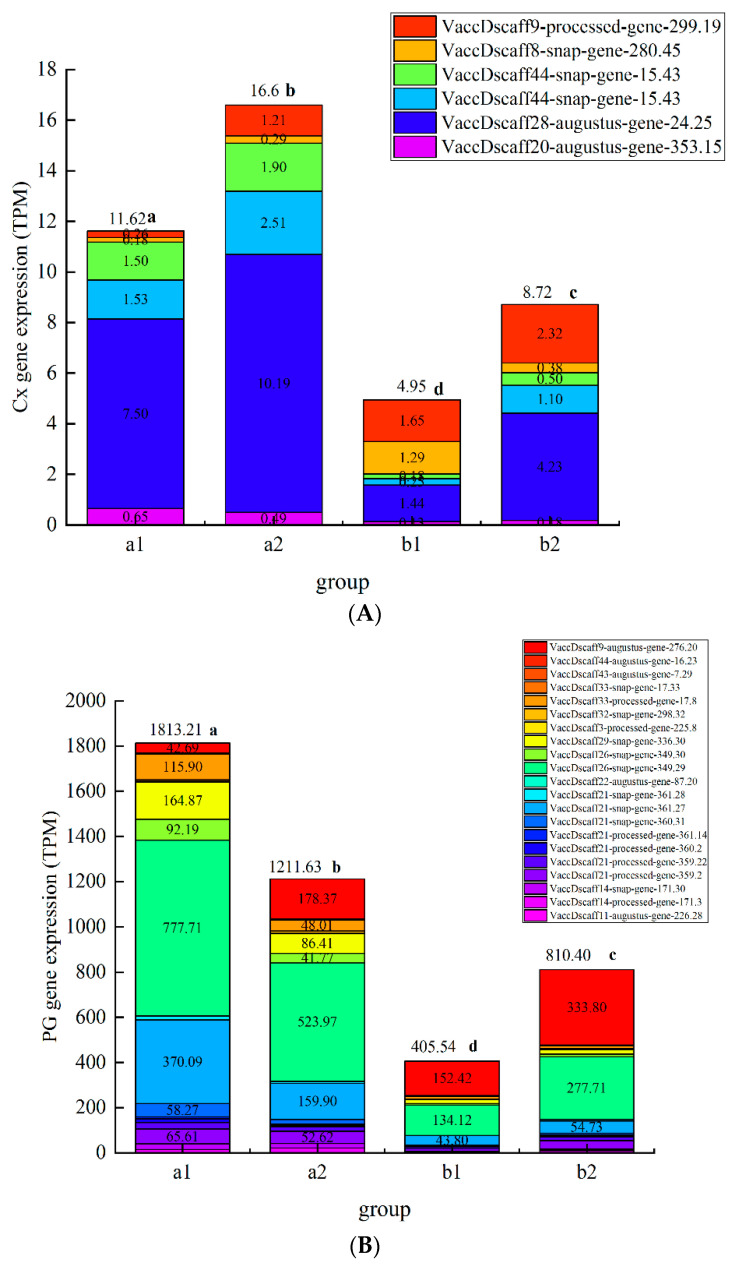
Plot map of gene expression a Plot map of PG gene expression, b Plot map of Cx gene expression, c Plot map of PME gene expression, d Plot map of β-Gal gene expression). (**A**): Plot map of polygalacturonase gene expression. (**B**): Plot map of cellulase gene expression. (**C**): Plot map of pectin methylesterase gene expression; (**D**): Map of β-galactosidase gene expression.

## Data Availability

All the data of this research are included in this manuscript.

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
