# Peer review of "Effect of Chitosan/Thyme Oil Coating and UV-C on the Softening and Ripening of Postharvest Blueberry Fruits"

_foods, 2022, doi:10.3390/foods11182795_

Round 1
Reviewer 1 Report
Experiments must be replicated at least three times for a research to be valid, which is not the case.
Even though formatting and presentation is less relevant than scientific content, the manuscript has many defects of presentation. Authors cannot expect that reviewers do their work when they do not take time to check author guidelines and other works published in the concerning journal.
I have made some comments until I realized that the experiments were not replicated or replication was not clearly stated, which is the same since a reviewer cannot assume what it is not written.
Keywords: I suggest adding “shelf-life” or “preservation” because it is a shelf-life study.
General: write citations and references as per author guidelines. English use need revision.
Line 37. Aging or antiaging?
Line 85. Incubator temperature?
Lines 99-103. This section is atypical. Equipment is usually described when cited in its respective place when methods are described.
Line 101. Balance description is not needed, it is a seldom provided detail in scientific papers.
Lines 114-118. Correct verb tense.
Lines 120-122. Describe the UV-C reactor (size, geometry, lamp location, sample amount and location with respect to lamp, model of lamp). How irradiation was measured? How much homogeneous was the illumination footprint?
Lines 126, 135. Do not encircle numbers.
Line 160, 166 and so on. Centrifugation conditions must be reported in multiples of “g”, i.e. acceleration of gravity. “rpm” units do not allow reproduce working conditions because centrifugation conditions depend on “rpm” and the radius of the circular movement.
In the formulas, if the symbols are written in lowercase letter in the formula, they must be written so in their definitions.
Line 208. What is tendency?
Figures must be self-sufficient, so acronyms must be defined in each one.
Line 234. This statement means that the experiments were not replicated, therefore, the study is not valid.
Author Response
Responses to comments for Review 1
Experiments must be replicated at least three times for a research to be valid, which is not the case.
Answer: Yes, we repeated the experiment three times, and the results are expressed as the mean±SD.
Even though formatting and presentation is less relevant than scientific content, the manuscript has many defects of presentation. Authors cannot expect that reviewers do their work when they do not take time to check author guidelines and other works published in the concerning journal.
Answer: We apologize for the inconvenience. We have checked this manuscript again to revise the defects of the presentation.
I have made some comments until I realized that the experiments were not replicated or replication was not clearly stated, which is the same since a reviewer cannot assume what it is not written.
Answer: We apologize for the confusion, and we did the replication. We have revised the inappropriate statement in the manuscript.
Keywords: I suggest adding “shelf-life” or “preservation” because it is a shelf-life study.
Answer: Thank you for the comments. “Preservation” was added to the keywords.
General: Write citations and references as per author guidelines. English use need revision.
Answer: Sorry, revision has been made for the citations and references according to the author guidelines.
Line 37. Aging or antiaging?
Answer: We apologize for the antiaging effect. The typo was revised in the manuscript.
Line 85. Incubator temperature?
Answer: 4℃. This information is aded to the manuscript.
Lines 99-103. This section is atypical. Equipment is usually described when cited in its respective place when methods are described.
Answer: Thank you, this section is amended.
Line 101. Balance description is not needed, it is a seldom provided detail in scientific papers.
Answer: this section is amended.
Lines 114-118. Correct verb tense.
Answer: Thank you, this section is amended.
Lines 120-122. Describe the UV-C reactor (size, geometry, lamp location, sample amount and location with respect to lamp, model of lamp). How irradiation was measured? How much homogeneous was the illumination footprint?
Answer: Thank you for the comments. Information was added.
Lines 126, 135. Do not encircle numbers.
Answer: Thank you, this section is amended.
Line 160, 166 and so on. Centrifugation conditions must be reported in multiples of “g”, i.e., acceleration of gravity. “rpm” units do not allow reproduce working conditions because centrifugation conditions depend on “rpm” and the radius of the circular movement.
Answer: Thank you, this section is amended.
In the formulas, if the symbols are written in lowercase letters in the formula, they must be written so in their definitions.
Answer: Thank you. This section has been amended.
Line 208. What is tendency?
Answer: We apologize for the typo. We have revised the manuscript.
Figures must be self-sufficient, so acronyms must be defined in each one.
Answer: Thank you. This section has been amended.
Line 234. This statement means that the experiments were not replicated; therefore, the study is not valid.
Answer: These sections were amended. We replicated the analysis, and the results are expressed as the mean±SD.
Reviewer 2 Report
Dear authors,
The results provides the basis for delaying postharvest softening and prolonging the shelf life of blueberries by chitosan/thyme oil coating and UV-C treatment. The effect of UV-C on destroying the molecular structure of DNA or RNA in microbial cells was described in the literature before. However, there are no relevant reports about the application of thyme essential oil in preserving blueberry coatings and molecular gene-level studies have not been reported up to now.
General comment:
· - Please explain the abbreviations when first mentioned (for example in line 16ff)
· - Avoid pseudo accuracy (for example line 340f)
Author Response
Responses to comments for Review 2
Dear authors,
The results provide the basis for delaying postharvest softening and prolonging the shelf life of blueberries by chitosan/thyme oil coating and UV-C treatment. The effect of UV-C on destroying the molecular structure of DNA or RNA in microbial cells has been described in the literature before. However, there are no relevant reports about the application of thyme essential oil in preserving blueberry coatings, and molecular gene-level studies have not been reported to date.
General comment:
- - Please explain the abbreviations when first mentioned (for example in line 16ff)
Answer: Thank you very much for the comments. The abbreviations were revised.
- - Avoid pseudo accuracy (for example line 340f)
Answer: Thank you; these were amended.
Reviewer 3 Report
The manuscript "Study of the chitosan / thyme oil coating and UV-C on the softening and ripening mechanism of postharvest blueberries" presents interesting research results. Storage protection of fruit and vegetables is a very important topic that can reduce storage losses in the food industry.
Detailed comments:
keywords - what does the word mechanism mean, what does it refer to?
The citations in the text do not comply with the journal's guidelines.
line 48 - use coatings instead of membranes.
line 48-53 - please add other coatings that can be used on blueberries, e.g. pullulan or pullulan with propolis extracts.
The authors were able to test the effect of the coating on the native blueberry microflora.
Figure 2, 3, 4, 5, 6, 7 - the drawings are not legible, they need to be enlarged.
Figure 8 - the caption is missing, the drawing is also not legible.
Chapter References - is not written according to the journals guidelines.
Author Response
Responses to comments for Review 3
The manuscript "Study of the chitosan/thyme oil coating and UV-C on the softening and ripening mechanism of postharvest blueberries" presents interesting research results. Storage protection of fruit and vegetables is a very important topic that can reduce storage losses in the food industry.
Detailed comments:
keywords - what does the word mechanism mean, what does it refer to?
Answer: We meant to refer to how the chitosan/thyme oil coating and UV-C work on the softening and ripening of postharvest blueberries. We have deleted this inaccurate keyword.
The citations in the text do not comply with the journal's guidelines.
Answer: The styles of citations have been amended.
line 48 - use coatings instead of membranes.
Answer: Thank you; this has been amended.
line 48-53 - please add other coatings that can be used on blueberries, e.g., pullulan or pullulan with propolis extracts.
Answer: we added some description as reviewer suggested.
The authors were able to test the effect of the coating on native blueberry microflora.
Answer: Thank you very much for the comments. We did not test the microflora in this manuscript. We will further test them and include the data in the next manuscript.
Figure 2, 3, 4, 5, 6, 7 - the drawings are not legible, they need to be enlarged.
Answer: Thank you. The figures have been amended.
Figure 8 - the caption is missing, the drawing is also not legible.
Answer: Thank you; the figure has been amended.
Chapter References - is not written according to the journals guidelines
Answer: Thank you. This section has been amended.
Round 2
Reviewer 3 Report
The corrections have been made in the text, I have no more comments.
Author Response
Thank you very much for your comments and work on this manuscript.
Sincerely
Jinjin Pei